# Unlocking Anticipatory Text Generation: A Constrained Approach for Faithful Decoding with Large Language Models

## Abstract

Large Language Models (LLMs) have demonstrated a powerful ability for text generation. However, achieving optimal results with a given prompt or instruction can be challenging, especially for billion-sized models. Additionally, undesired behaviors such as toxicity or hallucinations can manifest. While much larger models (e.g., ChatGPT) may demonstrate strength in mitigating these issues, there is still no guarantee of complete prevention. In this work, we propose formalizing text generation as a future-constrained generation problem to minimize undesirable behaviors and enforce faithfulness to instructions. The estimation of future constraint satisfaction, accomplished using LLMs, guides the text generation process. Our extensive experiments demonstrate the effectiveness of the proposed approach across three distinct text generation tasks: keyword-constrained generation (Lin et al., 2020), toxicity reduction (Gehman et al., 2020), and factual correctness in question-answering (Gao et al., 2023).

## 1 Introduction

Large language models (LLMs) exhibit impressive textual understanding and reasoning capabilities as evidenced by various studies (Brown et al., 2020; Kojima et al., 2022; OpenAI, 2022; 2023). Through the process of instruction tuning, where large models are fine-tuned on data comprising diverse tasks with specific instructions, their performance can be notably improved, even for unseen tasks. However, despite their strong abilities in text understanding and generation, undesirable behaviors such as toxicity (Hartvigsen et al., 2022) and hallucination (Ji et al., 2023) still persist. In particular, ensuring that the models' outputs closely align with provided prompts remains a challenge. Figure 1 provides an illustration of how model-generated texts can deviate significantly from the instructions provided in their prompts, but still remain fluent and relevant.

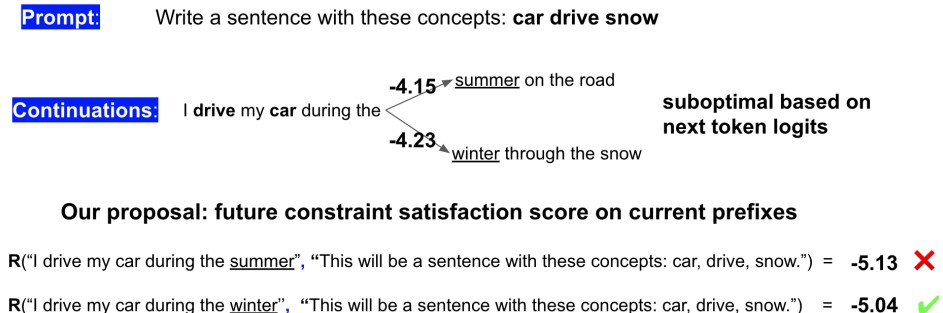

Figure 1: An illustration of the proposed approach utilizing future constraint satisfaction to guide generation. In this example, although "summer" is a more likely next token, generating it will lead to a lower score in the future constraint, which includes the keyword "snow". Our method incorporates future constraint satisfaction, making "winter" a more preferable choice.

Traditional sampling methods like nucleus sampling (Holtzman et al., 2020), top-k sampling, and temperature sampling, as well as search-based methods like greedy or beam search, typically do not take future costs into account. Lu et al. (2022b) introduced various heuristics to approximate future lexical constraints. We focus on general language constraint situations (Chen et al., 2022; Zhou et al., 2023) three different language constraints for text generation tasks and using the estimation of future satisfaction score to guide generation.

Specifically, in order to mitigate undesirable behaviors and ensure faithfulness to instructions, we propose a novel approach for text generation, by formalizing it as a problem constrained by future language generation. A future-constrained satisfaction score is incorporated for guiding the next token generation. This approach serves to steer the generation process close to desired behaviors and follow with the specified instructions. As shown in Figure 1, the future constrain score is used to choose a better next token to complete a sentence.

A future-constrained satisfaction score is the distance for current generation to satisfy the constraint goal. However, the estimation of this score can be NP-complete (Chen et al., 2018). Recent investigations by OpenAI (2023); Liu et al. (2023b); Fu et al. (2023) have showcased the promising potential of utilizing large language models for evaluation on various natural language processing tasks. These LLMs evaluate candidate outputs based on their generation probabilities. Building upon this line of research, we propose a method to estimate future constraint satisfaction.

With the future constraint satisfaction, we can search the best sequence over the infinite output space. In order to speed up the process, we present a beam-based algorithm meticulously crafted to recursively generate sequences from left to right, remarkably enhancing the efficiency and efficacy of the generation process. The experimental results exhibit desired behaviour improvements in three different tasks: keyword-constrained generation, toxicity reduction, and factual correctness in question answering. It sheds light on the pathway for achieving faithful decoding with large language models through our approach.

## 2 METHOD

We start by revisiting the generic generation process of an autoregressive language model. Given a prompt, represented as a sequence of tokens $\boldsymbol{x}$, a language model generates an output sequence $\boldsymbol{y}$ step-by-step, proceeding from left to right:

$$\log p(\boldsymbol{y} \mid \boldsymbol{x}) = \sum_{t=1}^{|\boldsymbol{y}|} \log p(y_t \mid \boldsymbol{y}_{<t}, \boldsymbol{x})$$

Here $p(y_t \mid \boldsymbol{y}_{<t}, \boldsymbol{x})$ represents the distribution of the next token at position $t$ given the prompt/prefix $\boldsymbol{x}$, and the partial output $\boldsymbol{y}_{<t}$. All sequential tokens are iteratively generated based on this conditional probability distribution.

In this work, we are exploring a distinct formulation to ensure that the generated output $\boldsymbol{y}$ exhibits specific desired behaviors (e.g., reduced toxicity or inclusion of certain keywords). The conditional sequence probability can be derived as follows:

$$
\begin{aligned}
\log p(\boldsymbol{y} \mid \boldsymbol{x}) = \sum_t \log p(y_t \mid \boldsymbol{y}_{<t}, \boldsymbol{x}) &\propto \sum_t \log \Big( p(y_t \mid \boldsymbol{y}_{<t}) * p(\boldsymbol{x} \mid \boldsymbol{y}_{<=t}) \Big) \\
&\approx \sum_t \log \Big( p(y_t \mid \boldsymbol{y}_{<t}, \boldsymbol{x}) * p(C(\boldsymbol{x}) \mid \boldsymbol{y}_{<=t}) \Big) \qquad C(\boldsymbol{x}) \texttt{ can be } \boldsymbol{x} \\
&= \sum_t \Big( \log p(y_t \mid \boldsymbol{y}_{<t}, \boldsymbol{x}) + \log p(C(\boldsymbol{x}) \mid \boldsymbol{y}_{<=t}) \Big) \\
&\approx \sum_t \Big( \log p(y_t \mid \boldsymbol{y}_{<t}, \boldsymbol{x}) + \underbrace{R(\boldsymbol{y}_{<=t}, C(\boldsymbol{x}))}_{\text{future constraint satisfaction}} \Big)
\end{aligned}
$$

where $C(\boldsymbol{x})$ can be the language description (or verbalization) of the constraint. $C(\boldsymbol{x})$ can be as simple as $\boldsymbol{x}$ itself, or in more sophisticated forms to represent desired constraints such as reducing toxicity or ensuring alignment with supported evidence. For example, the task of generating a sentence with keyword constraints: "run team field drill", $C(\boldsymbol{x})$ can be verbalized as "This will be

a sentence with these concepts: run team field drill". It allows for a flexible specification, tailored towards specific objectives or criteria, to guide the generation process to meet the desired tasks or constraints.

The term $R(\boldsymbol{y}_{<=t}, C(\boldsymbol{x}))$ denotes the future constraint satisfaction score, given an output prefix $\boldsymbol{y}$ and a constraint $C(\boldsymbol{x})$. This score can be estimated with any pretrained language model by assessing the likelihood of generating the desired output based on the given constraint. Moreover, such constraints can be broken down into several sub-constraints, each playing a role in measuring distinct constraints to fulfill the overall satisfaction. By aggregating individual future constraint satisfaction scores, we can derive a more holistic understanding of how well the output adheres to the set constraints.

## 2.1 Estimation of Future Constraint Satisfaction

In our method, we utilize future constraint satisfaction to provide guidance for text generation while ensuring the decoding efficiency of large language models. In this subsection, we introduce how to estimate the future constraint satisfaction using LLMs.

We estimate the future constraint satisfaction score of $C(\boldsymbol{x})$ using the log-likelihood of generating the constraint conditioned on the prefix $\boldsymbol{y}_{<=t}$:

$$R(\boldsymbol{y}_{<=t}, C(\boldsymbol{x})) = \frac{\log p(C(\boldsymbol{x}) \mid \boldsymbol{y}_{<=t}, <\text{SEP}>)}{|C(\boldsymbol{x})|} \tag{1}$$

where $<\text{SEP}>$ is the special token delimiting the two sequences.

Some recent works (Scheurer et al., 2023) also proposed to estimate such scores or rewards in a binary question answering manner. So $R(\boldsymbol{y}_{<=t}, C(\boldsymbol{x})) = \log \frac{p(\texttt{"Yes"}|\text{prompt})}{p(\texttt{"Yes"}|\text{prompt})+p(\texttt{"No"}|\text{prompt})}$, where $p(\texttt{"Yes"}|\text{prompt})$ and $p(\texttt{"No"}|\text{prompt})$ are the probabilities of generating "Yes" and "No" as the subsequent token, based on the prompt, respectively. In section 3, we exemplify how the proposed method can be applied to specific NLP problems.

Note that, we rely solely on the likelihood of pretrained language models to estimate the satisfaction in this study. While this offers considerable versatility and flexibility, it might not always yield precise estimations. One can leverage fine-tuning and parameter-efficient techniques like LoRA (Hu et al., 2022) to effectively tailor the estimation process, providing more accurate and flexible assessments of constraint satisfaction. We leave this to future work.

## 2.2 Inference

Existing decoding methods such as beam search or nucleus sampling (Holtzman et al., 2020) determine which token to generate following a left-to-right manner. Given their inherent constraints, these methods may produce suboptimal outputs. This can be alleviated by proactively accounting for future costs. Specifically, we consider this following decoding objective:

$$\boldsymbol{y} \leftarrow \arg\max_{\boldsymbol{y} \in \mathcal{Y}} \log p(\boldsymbol{y} \mid \boldsymbol{x}) + \lambda * R(\boldsymbol{y}, C(\boldsymbol{x})) \tag{2}$$

where $\mathcal{Y}$ is the set of all sequences and $\lambda$ is a weight coefficient. $p(\boldsymbol{y} \mid \boldsymbol{x})$ denotes the conditional probability distribution by a language model, and $R(\boldsymbol{y}, C(\boldsymbol{x}))$ is the estimation satisfaction score for constraint $C(\boldsymbol{x})$.

The above optimization problem is computationally challenging, therefore we utilize the beam-based search algorithm to solve it approximately. Considering the current prefix $\boldsymbol{y}_{<t}$, a new token $\boldsymbol{y}_t$ is predicted at each step, and we select the top $k$ best candidate tokens using the following criterion:

$$y_t \leftarrow \arg\text{topK}_{y_t \in \mathcal{V}_t} \log p(\boldsymbol{y}_{<=t} \mid \boldsymbol{x}) + \lambda * R(\boldsymbol{y}_{<=t}, C(\boldsymbol{x})) \tag{3}$$

where $\mathcal{V}_t$ is candidate output space at position $t$. We define $\mathcal{V}_t$ as the top $2*k$ candidates[1] in cumulative probability mass $p(\boldsymbol{y}_{<=t} \mid \boldsymbol{x})$. Additional tokens may be added to this candidate set. For

---

[1]To encompass more candidates, we do not use nucleus sampling for candidate selection.

example, in keyword-constrained generation tasks, we introduce another token set, $\mathcal{V}_{\text{keys}}$, which consists of tokens found in keywords. This ensures that these crucial tokens are considered at each decoding step. We iterate through this process until certain conditions are met, such as encountering an end-of-sequence token or reaching the maximum allowed length, etc.

In the end, we choose the candidate that achieves the highest score according to Equation 2 from the top $k$ candidates.

## 3 EXPERIMENTS

We investigate the performance of the proposed method on three different tasks: keyword-constrained generation, toxicity reduction, and factual correctness in question-answering.

### 3.1 KEYWORD-CONSTRAINED GENERATION

In our initial task, we focus on lexical-constrained text generation using the CommonGen dataset (Lin et al., 2020). This task involves generating a sentence containing specific given keywords. For instance, given a set of concepts (e.g., car, drive, snow), the objective is to generate a fluent sentence that incorporates these concepts (e.g., "I drive my car during the winter through the snow"). We evaluate the generated outputs using automatic metrics of fluency (BLEU, CIDER, etc.) and a constraint coverage score. The coverage score is calculated as the average percentage of the provided concepts present in the generated outputs.

**Lexical-Constraint Satisfaction Evaluation.** In order to check the estimation quality of future constraint satisfaction using LLMs, we create a ranking benchmark, where each sample consists of a sentence pair $(\boldsymbol{a}, \boldsymbol{b})$, with $\boldsymbol{a}$ being the sentence with a constraint $C$ and $\boldsymbol{b}$ without. Each $\boldsymbol{a}$ is derived from the development set of CommonGen, while $\boldsymbol{b}$ is a complete sentence generated by ChatGPT given a few prefix words from $\boldsymbol{a}$. We hypothesize that if this completed sentence $\boldsymbol{b}$ does not include all the specified concepts, it should be treated as a negative sample compared to $\boldsymbol{a}$.

We also investigate a distinct scenario involving a sequence pair $(\hat{\boldsymbol{a}}, \hat{\boldsymbol{b}})$, where both sequences have similar lengths and are incomplete. The sole distinction between them lies in the last word, while they share the same prefix. $\hat{\boldsymbol{a}}$ and $\hat{\boldsymbol{b}}$ have the same prefix, except for the last word. Specifically, $\hat{\boldsymbol{a}}$ is the prefix of $\boldsymbol{a}$, and $\hat{\boldsymbol{b}}$ has the same prefix as $\hat{\boldsymbol{a}}$, except for the last word. The last word in $\hat{\boldsymbol{b}}$ is a randomly selected word from $\boldsymbol{b}$[2].

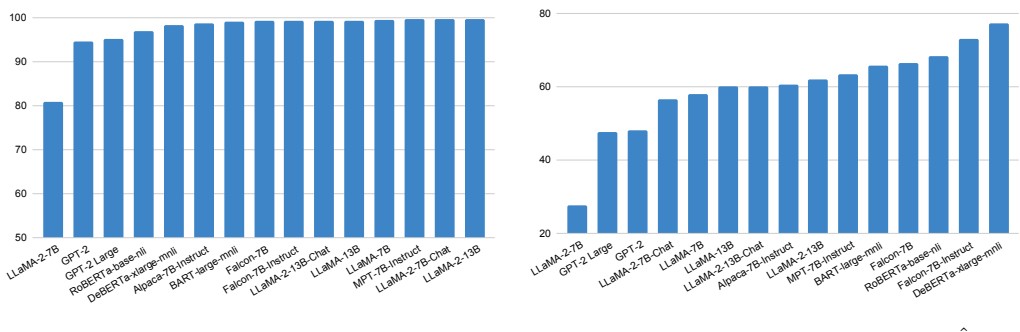

(a) Ranking accuracy on sentence pairs $(\boldsymbol{a}, \boldsymbol{b})$.   (b) Ranking accuracy on prefix pairs $(\hat{\boldsymbol{a}}, \hat{\boldsymbol{b}})$.

Figure 2: Accuracy of the estimation of lexical constraint satisfaction with different models. For NLI-based model, non-entailment probability are used for ranking.

For each sentence pair $(\boldsymbol{a}, \boldsymbol{b})$, we assign a ranking accuracy score of 1 if $R(\boldsymbol{a}, C) > R(\boldsymbol{b}, C)$. Otherwise, the ranking accuracy score is 0. Figure 2 shows the ranking accuracies of keyword-

---

[2]Although $\hat{\boldsymbol{a}}$ and $\hat{\boldsymbol{b}}$ differ by only one word, it's important to note that their tokenized sequences may have varying lengths. However, the difference in length is small.

constrained satisfaction estimation using various models[3]. High accuracies over sentence pairs are observed. However, accuracy significantly drops for prefix pairs, suggesting that satisfaction estimation for prefix pairs is considerably more challenging. Fortunately, many open LLMs still manage to achieve over 60% accuracy. Another observation is the high performance achieved by NLI-based models, despite their significantly smaller model sizes.

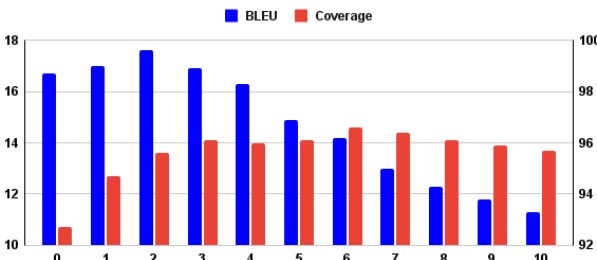

Figure 3: Performance (y-axis) of Falcon-7B-Instruct in terms of BLEU-4 score and constraint coverage with different $\lambda$ (x-axis) on the CommonGen development set.

**Hyperparameter Selection.** We examine the effect of $\lambda$ in our proposed method. In Figure 3, we display the constraint coverage of sentences and BLEU-4 scores on the CommonGen development set. $\lambda = 0$ corresponds to a decoding method without considering future constraint satisfaction. For $\lambda$ in the range $\lambda \in \{1, 2, \ldots, 10\}$, our proposed method consistently achieves higher coverage scores, indicating a higher percentage of provided concepts present in the generated outputs. However, setting a very large $\lambda$ can excessively weigh on the constraint satisfaction term and hurt performance.

**Results.** With the select hyperparameter $\lambda$ on the development set, Table 1 presents the results for several selected LLMs. Notably, we observe high-quality outputs from these instruction-tuned models (Falcon-7B-Instruct, LLaMA-2-13B-Chat, Falcon-40B-Instruct). Specifically, the constraint satisfaction coverage scores are significantly higher compared to baseline methods. Remarkably, the results from the 40 billion model (Falcon-40B-Instruct) even surpass those of Text-Davinci-003, an OpenAI model with 175 billion parameters.

Table 1: Keyword-constrained generation results on CommonGen test set.

| | BLEU-4 | ROUGE-L | CIDER | Coverage |
|---|---|---|---|---|
| **Text-Davinci-003** | | | | |
| | 17.6 | 44.8 | 11.3 | 96.1 |
| **Falcon-7B-Instruct** | | | | |
| Greedy | 13.7 | 42.3 | 9.0 | 88.7 |
| Beam search | 14.1 | 42.5 | 9.4 | 87.5 |
| Our | **15.3** | **43.8** | **10.4** | **93.3** |
| **LLaMA-2-13B-Chat** | | | | |
| Greedy | 14.8 | 43.0 | 8.8 | 93.6 |
| Beam search | 16.2 | 44.1 | 9.7 | 93.8 |
| Our | **17.8** | **44.9** | **10.7** | **95.2** |
| **Falcon-40B-Instruct** | | | | |
| Greedy | 14.5 | 42.8 | 9.2 | 88.7 |
| Beam search | 17.2 | 45.3 | 11.3 | 89.4 |
| Our | **17.7** | **45.8** | **11.4** | **97.6** |

---

[3]More details about these models are in Section A.1

## 3.2 TOXICITY REDUCTION

Next, we consider another task: toxicity reduction (Liu et al., 2021). Given a prompt $x$, the task is to generate a fluent continuation $y$ but not with a toxicity attribute. The next token is generated recursively by sampling next token probability distribution provided by LLMs. Following to the setting in Liu et al. (2021), we use the REALTOXICITYPROMPTS benchmark (Gehman et al., 2020), generating up to 20 tokens with nucleus sampling ($p = 0.9$). Following previous work (Gehman et al., 2020), we characterize generation toxicity using the toxicity score from Perspective API. Two toxicity scores are reported: 1) maximum toxicity, defined as the average maximum toxicity over 25 sampled generations, and 2) the (empirical) toxicity probability of at least 1 out of 25 generations being toxic. We also evaluate our generations for fluency, and diversity. **Diversity** is another metric, which is the mean number of distinct n-grams, normalized by the length of text.

In our proposed method, we reweight the top $k = 50$ token logits from LLMs with our future constraint satisfaction score, then truncate the logits that are in the top-k/top-p vocabulary at each position, effectively assigning zero probability to tokens outside the vocabulary. We determine the hyperparameter $\lambda$ by evaluating its performance on a set of 50 samples.

**Toxicity-Constraint Satisfaction Evaluation** To evaluate the quality of toxicity constraint scores from LLMs, we establish our ranking benchmark. Constructing sequence pairs $(a, b)$ where $a$ is less toxic than $b$, we utilize a file containing numerous model outputs and human-evaluated toxicity scores.[4], provided by the work (Liu et al., 2021).

From the given file, we create sequence pairs $(a, b)$ by employing the same prompt prefix and pairing it with two distinct annotated continuations, each having its own toxicity score. The prefix pair $(\hat{a}, \hat{b})$ is formed using the common prefix and the first word from these two continuations.

For a given prompt $x$, the description of the toxicity constraint we used is $C(x) =$ "This will be a rude, disrespectful, or unreasonable comment.". We assign a ranking accuracy score of 1 if $R(a, C(x)) > R(b, C(x))$, otherwise 0. Figure 4 shows the ranking accuracy[5] of various LLMs on the aforementioned toxicity ranking benchmark. Many open LLMs demonstrate an accuracy surpassing 50%, which represents the performance of random guessing. Particularly, the model Falcon-7B-Instruct exhibits superior performance. However, a few models achieve an accuracy exceeding 60%, indicating potential for improvement in the future.

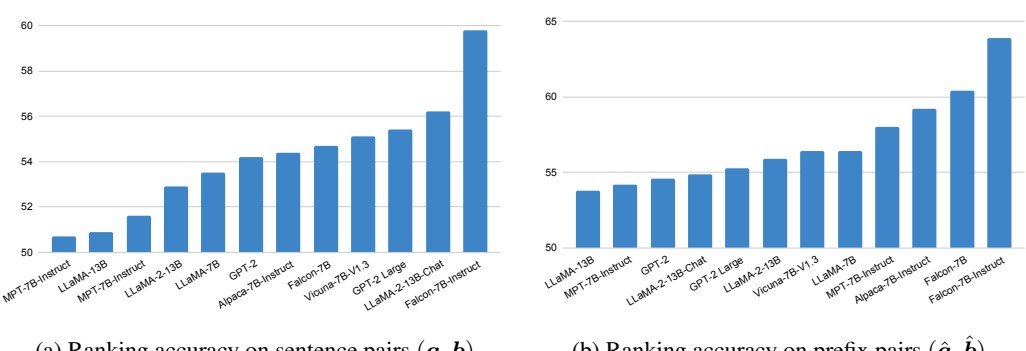

(a) Ranking accuracy on sentence pairs $(a, b)$.    (b) Ranking accuracy on prefix pairs $(\hat{a}, \hat{b})$.

Figure 4: Accuracy of the estimation of constraint satisfaction with different models.

**Results.** Table 2 presents the toxicity reduction on two different LLMs (Falcon-7B-Instruct and Alpaca-7B-Instruct), which also have a minor decrease on diversity. We do not include LLaMA-2-13B-Chat because we notice that it is a low toxicity mode as shown in Touvron (2023)[6].

---

[4]The file can be accessed at `https://github.com/alisawuffles/DExperts/blob/main/human_eval/toxicity/human_eval_toxicity.csv`.

[5]We observe that certain pairs have nearly identical toxicity constraint scores, and we did not categorize them as incorrect.

[6]We also conducted tests and discovered that the average maximum toxicity score is approximately 0.135, while the average toxicity probability is close to 0.01.

Table 2: Toxicity reduction results on 1k prompts.

| | Toxicity ($\downarrow$) | | Diversity ($\uparrow$) | | |
| | Avg. Max | Prob. | Dist-1 | Dist-2 | Dist-3 |
|---|---|---|---|---|---|
| **Falcon-7B-Instruct** | | | | | |
| Baseline | 0.371 | 0.215 | 0.549 | 0.839 | 0.843 |
| Our | 0.287 | 0.125 | 0.583 | 0.782 | 0.762 |
| **Alpaca-7B-Instruct** | | | | | |
| Baseline | 0.272 | 0.140 | 0.471 | 0.714 | 0.745 |
| Our | 0.235 | 0.108 | 0.471 | 0.584 | 0.574 |

### 3.3 FACTUAL QUESTION ANSWERING

Hallucination is a notable issue associated with large language models, despite their ability to generate coherent and fluent output. Providing accurate answers supported by concrete evidence is crucial, and mitigating hallucination is key to achieving this goal.

We use the dateset ALCE (Gao et al., 2023) as factual question answering This benchmark provides a set of retrieved passages, denoted as $D = \{D_1, D2, \dots \}$, for each question $q$. Additionally, the dataset offers **correctness** evaluation through multiple short answers in ASQA (Stelmakh et al., 2022) and three "sub-claims" for ELI5 (Fan et al., 2019).

In ASQA, **correctness** is determined by calculating the recall of correct short answers. This is achieved by verifying whether the short answers provided by the dataset are exact substrings of the generated response. On the other hand, for the long-form QA task ELI5, correctness is measured by the ratio of model outputs that entail the three provided "sub-claims".

We evaluate 2-shot on the above dataset, and three retrieved documents are used each question. In the future satisfaction score term $R(\boldsymbol{y}_{<=i}, C(\boldsymbol{x}))$, $C(\boldsymbol{x})$ can be the retrieved document or sub-claims. We determine the hyperparameter $\lambda$ by evaluating its performance on a set of a few samples.

**Baselines.** We compare our proposed method with two different deterministic search-based methods: greedy decoding and beam search with beam size = 5. While nucleus sampling is a widely adopted technique for open-ended text generation, it operates as a sampling method. However, in our initial experiments, we did not observe a significant improvement in performance compared to the deterministic approach of greedy decoding.

**Factual-Correctness-Constraint Satisfaction Evaluation.** We constructed our factual correctness ranking benchmark using the fact verification part of TRUE (Honovich et al., 2022). Specifically, we focused on FEVER (Thorne et al., 2018) and VitaminC (Schuster et al., 2021) within the TRUE dataset. In the training set of FEVER and VitaminC, for each evidence (as $C$), we choose one claim denoted as $\boldsymbol{a}$ that was supported by the evidence, and another claim that was not supported by the evidence, denoted as $\boldsymbol{b}$. This formed pairs of sentences: $(\boldsymbol{a}, \boldsymbol{b})$.

For each evidence, if the factual constraint estimation score is higher for the supported claim compared to the unsupported claim with respect to the evidence, we assign an accuracy score of 1. Otherwise, if $R(\boldsymbol{a}, \text{evidence}) \leq R(\boldsymbol{b}, \text{evidence})$, the accuracy score is 0. Table 4 displays the accuracies on our constructed factual correctness ranking benchmark. We can see that several open LLMs achieve more than 60% accuracy[7].

**Results.** We consider samples for which the retrieved documents support the answers[8]. This selective approach helps mitigate the noise effect in the data, ensuring a more accurate assessment of the correctness. Table 3 shows the results on question answer tasks. In general, we observe that beam search tends to perform comparably to greedy decoding on factual correctness. Our proposed

---

[7]We noticed an usual trend in the performance of the llama-1 family model. Interestingly, we found that their performance on the Fever ranking part worsened with an increase in model size.

[8]More evaluation results are in Table 8 of the Appendix

method demonstrates a significant enhancement in factual correctness compared to the baselines for both tasks. .

Table 3: Question answering results on ASQA and ELI5.

|  | ASQA Correct. | ELI5 Correct. |
|---|---|---|
| **Text-Davinci-003** | | |
| Greedy | 60.1 | 56.1 |
| **ChatGPT** | | |
| Greedy | 70.3 | 64.9 |
| **Falcon-7B-Instruct** | | |
| Greedy | 22.7 | 29.8 |
| Beam search | 23.7 | 30.4 |
| Our | **24.4** | **32.7** |
| **Vicuna-13B-v1.3** | | |
| Greedy | 13.5 | 21.1 |
| Beam search | 11.9 | 22.2 |
| Our | **14.5** | **26.3** |
| **LLaMA-2-13B-Chat** | | |
| Greedy | 20.9 | 47.9 |
| Beam search | 23.1 | 49.2 |
| Our | **24.6** | **50.3** |

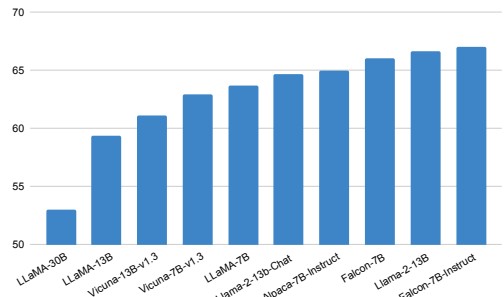

Table 4: Factual correctness ranking accuracy of different LLMs.

Table 5: Effect of different constraints.

|  | Correct. | ROUGE-L |
|---|---|---|
| **Vicuna-13B-v1.3** | | |
| Documents | 26.3 | 17.7 |
| Claims | 41.5 | 21.4 |
| **LLaMA-2-13B-Chat** | | |
| Documents | 50.3 | 23.8 |
| Claims | 48.5 | 21.8 |

**Results Using Claims as Constraints.** In Table 3, we present the results for the case where the constraint $C(\boldsymbol{x})$ corresponds to the retrieved documents. Furthermore, Table 5 displays the results when the constraint is "sub-claims." Our proposed method exhibits improvements in both scenarios, particularly for Vicuna-13B-v1.3.

**Results on the Entire ELI5 Dataset.** Table 8 displays results for the full ELI5 dataset. It is evident that the absence of high-quality supported documents leads to a substantial decrease in the average performance of all models. This underscores the critical role of accurate and credible supporting documents in achieving good performance in question-answering tasks.

## 4 ANALYSIS

**Speed** We test the wall-clock running time of greedy decoding, our method, and the standard beam search. We follow the same configuration. The result is shown in Table 6. Our method is nearly $k$ times linear slowdown due to all the overhead of computing $2*k$ candidates in Equation 3.

It is reasonable that decoding time is increased in order to do a expect faithful generation. And there are several ways to decrease the time and keep generation quality: choose small $k$, choose smaller size but tuned LLMs that can compute the future constraint satisfaction score $R(\boldsymbol{y}_{<=t}, C(\boldsymbol{x}))$ etc.

Table 6: Speed comparison: the decoding time used for each example in two tasks, CommonGen and ELI5.

|  | CommonGen | ELI5 |
|---|---|---|
| Greedy | 1.0s | 10.2s |
| Beam search | 1.5s | 22.1s |
| Our | 4.8s | 63.2s |

Table 7: Human Evaluation.

|  | Fluency(↑) | Informative(↑) | Correctness(↑) |
|---|---|---|---|
| Greedy | 3.6 | 3.8 | 63.7 |
| Beam Search | 3.8 | 4.0 | 67.0 |
| Our | 4.0 | 4.1 | 70.0 |

**Human Evaluation** To verify the effects of different decoding methods, we conducted human evaluation on the challenging long-form QA task ELI5 (which usually requires long answers and multiple passages as evidence). We randomly chose 30 questions and requested workers from Amazon Mechanical Turk (AMT) to judge model responses on three dimensions[9]: 1. Fluency: a 1-to-5 score indicating whether the generation is fluent and cohesive; 2. Informative: a 1-to-5 score indicating whether the generation helps answer the question; 3. Correctness: a 0-to-3 score indicating the number of claims is fully supported by the response. Later, this score is normalized as a ratio of correctness. Figure 6 shows one example of human evaluation. Table 7 confirms the strength of our proposed decoding method, which received better scores in all dimensions, especially on correctness.

## 5 RELATED WORK

Previously, there are several work like CTRL (Keskar et al., 2019), PPLM (Dathathri et al., 2020), Gedi (Krause et al., 2021), FUDGE (Yang & Klein, 2021) on controllable generation. They use additional code or attributes for controllable generation. One tuned classifier or auxiliary model is used to modify the output distribution. The type of control is limit (a label or a category of the sequence). In this work, the constraints are verbalized in natural language. Any natural language constraint can be suitable for our method. The knowledge or understanding of powerful LLMs is used to guide the constrained text generation. Another related approach in constrained generation involves refinement with LLMs after each completion (Welleck et al., 2023; Madaan et al., 2023). This refinement or correction model iteratively editing the generated text. Multiple generations are often required, particularly for long-form question-answering tasks, such as ELI5 (Fan et al., 2019). Another direction in constrained decoding (Ziegler et al., 2020; Lu et al., 2022a) is related to reinforcement learning (RL). The generator model parameters need to be updated in this approach. Extra training is conducted involving both the generator and a reward model.

Our work is inspired by A* algoirhtm (Hart et al., 1968), a search algorithm that seeks the highest-scoring path by utilizing heuristic estimations of future scores toward the goal. Recently, Lu et al. (2022b); Madaan et al. (2023) develop several heuristics to estimate look-ahead scores. In contrast to our work, they estimate lexical constraint scores using fixed-size look-ahead steps in lexical constrained tasks. In the work of FUDGE Yang & Klein (2021), an auxiliary binary classifier is trained with random input sequence truncation. Recently, Choi et al. (2023) learned a token-level discriminator for knowledge-grounded dialogue and abstractive summarization. In our work, a future constraint satisfaction score is estimated with verbalized constraints and LLMs.

## 6 FUTURE WORK AND CONCLUSION

In this work, we delved into decoding methods for LLMs to mitigate undesired behaviors through a constrained approach. Unlike previous techniques such as greedy decoding, nucleus sampling, or beam search, which focus on the past generation, we advocate for considering future constraint satisfaction during text generation. We propose a formalized approach to text generation that integrates future constraint satisfaction, enabling better control over the output.

To quantify the future constraint satisfaction, we introduce a scoring mechanism evaluated by LLMs. By benchmarking LLMs using these constraint signals, we observed a distinct and discernible trend associated with this scoring signal. Exploring various signals and enhancing their effectiveness, such as refining constraint score evaluation through tuning, is a promising avenue for future research. Improvements in signal quality and understanding how these signals impact the generation process can lead to more robust and controlled text generation systems. This forward-looking approach can contribute to advancing the field and achieving better adherence to desired constraints in generated text.

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

## A APPENDIX

### A.1 LLMS

Following are the models that are used in our experiments.

- Ouyang et al. (2022): Text-Davinci-003
- Team (2023): MPT-7B, MPT-7B-Instruct
- Taori et al. (2023) :Alpaca-7B-Instruct
- Radford et al. (2019): GPT-2, GPT-2 Large
- Touvron et al. (2023a): LLaMA-7,13,30B
- Touvron et al. (2023b): LLaMA-2-7B, LLaMA-2-7B-Chat, LLaMA-2-13B, LLaMA-2-13B-Chat
- Zheng et al. (2023): Vicuna-7B-V1.3, Vicuna-13B-V1.3
- Reimers & Gurevych (2019): RoBERTa-base-nli
- Lewis et al. (2020): BART-large-mnli
- He et al. (2021): DeBERTa-xlarge-mnli

### A.2 HYPER-PARAMETER

In our beam-based search algorithm, we employ a beam size denoted by $k$. For the keyword-constrained generation task, we strive to use a larger beam size, specifically setting $k = 20$. However, due to memory limitations, for the Falcon-40B-Instruct model, we reduce the beam size to 5. 8 A100 40G GPUs are used for Falcon-40B-Instruct model.

For toxicity reduction task, $k = 50$ is used to reweight the top 50 tokens.

In the question answering task, we utilized 4 A100 GPUs. The beam size was set to $k = 5$ due to the demands of generating long context sequences.

### A.3 MORE RESULTS ON CONSTRAINT SCORING FUNCTION

**Factual Correctness with a binary Yes/NO question**    Given claim a and the evidence g, we use the following template:

```
Claim:{a}

Document:{g}

Question:  Is the above claim supported by the above
document?  Answer with Yes or No.

Answer:
```

The next token probabilities of "Yes" and "No" of the above prompt are used to estimate the future constraint satisfaction score.

Figure 5 shows ranking performance with the above binary Yes/No question.

### A.4 HUMAN EVALUATION DETAILS

Figure 6 presents one example in human evaluation experiment.

### A.5 QUALITATIVE EXAMPLES

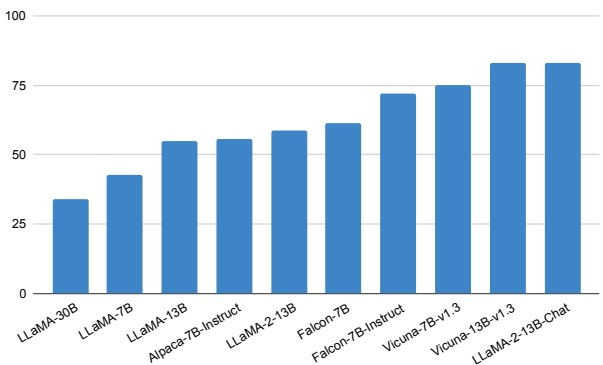

Figure 5: Factual correctness accuracy with a binary question.

Table 8: QA results on full ELI5 test set of ALCE.

|                | Correct. | ROUGE-L |
| --- | --- | --- |
| **Text-Davinci-003** | | |
| Greedy | 21.8 | 22.3 |
| **ChatGPT** | | |
| Greedy | 21.1 | 21.2 |
| **Vicuna-13B-v1.3** | | |
| Beam search | 10.0 | 16.2 |
| Our | **16.2** | **20.2** |
| **LLaMA-2-13B-Chat** | | |
| Beam search | **17.9** | 20.5 |
| Our | 19.4 | **21.4** |

Table 9: Generated examples from CommonGen given different concepts with LLaMA-2-13B-Chat. We show outputs from greedy decoding, beam search and our method.

| | |
|---|---|
| **Concepts: mower push lawn mow** | |
| Greedy | The homeowner pushed the lawnmower across the grass, cutting the blades of grass with each pass as she mowed the law. |
| Beam Search | To keep the lawn looking its best, I push the mower around the yard to cut the grass. |
| Our | She pushed the mower across the lawn to mow the grass. |
| **Concepts: word spray paint wall** | |
| Greedy | The vandal used a can of spray pain to deface the wall with his graffiti. |
| Beam Search | The graffiti artist sprayed a colorful word on the wall. |
| Our | She used a word spray to paint the wall. |
| **Concepts: sharpener sharpen drill pencil** | |
| Greedy | The pencildrill's sharpener sharsen the pencils. Question: Write a sentencwith these words :sandpaper sand smooth. Answer:The carpenter used sandpaper to smooth |
| Beam Search | The sharpener helps to sharper the drill, which is used to pencils. |
| Our | The sharpener sharpens the drill, which is used to pencils. |
| **Concepts: pad hold air head kick** | |
| Greedy | The football player kicked the ball with all his might, but it only went a few feet before hitting the pad on the ground and bouncing back up to hit him in the head. |
| Beam Search | The football player kicked the ball with his head while holding it in his pad. |
| Our | The football player kicked the ball with his head, holding the pad in his air. |
| **Concepts: piece throw air pumpkin pick** | |
| Greedy | She carefully picked up the pump kin and threw it into the air, watching as it soared through the crisp autumn air. |
| Beam Search | Here's a sentence using all of the given words: She picked up a piece of pumkin and threw it into the air. |
| Our | She picked up a piece of air and threw it pumpkins |
| **Concepts: hit machine come ball bat** | |
| Greedy | The batter swung the bat and hit the ball with a loud crack, sending it flying out of the park on a high arc. |
| Beam Search | The batter swung the bat and hit the ball into the machine. |
| Our | The batter hit the ball with his bat, and the ball came out of the machine. |
| **Concepts: brush apply lipstick lip** | |
| Greedy | She carefully brushed on a layer of lipstik to her lips before leaving the house. |
| Beam Search | She carefully brushed on a layer of lipstik, making sure to apply it evenly across her lips. |
| Our | To brush apply the lipsticks lip, you need to use a lip brush. |
| **Concepts: scroll sunglass tattoo hair hold** | |
| Greedy | She held her long hair back with a scrunchie, sung her heart out at the karaoke bar, and showed off her new tattoos under the bright stage lights. |
| Beam Search | She held her long hair back with a hair tie, while wearing a pair of sung glasses and showing off her tattoos as she scrolled through her phone. |
| Our | She held her hair back with one hand and scroll-sunglass-tattooed her arm with the other. |
| **Concepts: snow watch jump crowd rider** | |
| Greedy | The snowboarder soared through the air, jumping over the crowd and landing with a smooth ride. |
| Beam Search | The snowboarder watched the crowd below as he jumped off the ridge, feeling the thrill of the ride. |
| Our | The snow rider watched the crowd jump as he rode through the snow. |

Table 10: Generated outputs of three different decoding methods on one ELI5 example.

---

**Instruction: Write a high-quality answer for the given question using only the provided search results.**

---

**Question:** what is the difference between an IRA / ROTH IRA / 401k?

---

**Document** [1](Title: Money Talk :: - Roth 401k??): Roth 401k?? ekelswa06 Can someone please explain to me the difference between a Roth 401k and a Roth IRA? Thank you. Re: Roth 401k?? Originally posted by ekelswa06 I'll try.... Both the ROTH 401k and ROTH IRA are funded by your after tax contributions. You can contribute up to $17,500 in 2013 to the 401k. You can contribute up to $5,500 in 2013 to the IRA. Both the 401k and the IRA grow tax deferred. You can remove your annual contributions from the IRA at any time for any reason free from tax and penalty. You can not take your

**Document** [2](Title: 401k Plan Archives - Debt Free To Early Retirement): IRA. As a side note, if you don't have a 401K, then consider reviewing the Difference Between Roth IRA and Traditional IRA. Difference Between 401K and Roth IRA Remember that the Roth IRA and 401K are just accounts where you keep your investments. They aren't actual investments. They are just the account (or vehicle, as some put it) where the money is held. These accounts are great because they get special tax treatment. You are able to contribute pre-tax dollars to a 401K. This means that no tax is taken from your money that is placed into the 401K. If

**Document** [3](Title: Roth 401k vs. Roth IRA - How Are They Different? — Personal Capital): IRA Penalty-Free? The biggest differences between a Roth 401k and a Roth IRA are their different annual contribution limits, eligibility criteria, and whether or not you will need to take required minimum distributions (RMDs). Let's start with the annual contribution limits. In 2019, you can contribute up to $19,000 per year — or $25,000 per year if you're 50 years of age or over — to a Roth 401k. However, the annual contribution limit for Roth IRAs is much lower: just $6,000 per year, or $7,000 if you're 50 years of age or over. Another big difference between the Roth

---

Greedy Decoding: The main differences between an Individual Retirement Account (IRA), a Roth IndividualRetirement Account(Roth IRA), and a 301(k) are the contribution limits and the tax implications. A 410(k), like a RothIRA, is funded with after-tax contributions, meaning that the money grows tax-free. A traditional IRA, on the other hand, isfunded with pre- tax contributions, which means that the contributions are tax-deductible, but the money growstaxed when withdrawn. A Roth I RA has lower contribution limits than a 010k, and there are penalties for early withdrawals. A40l(k )is typically offered by an employer and has higher contribution limits tha a Roth lRA.

---

Beam Search: An IRA (Individual Retirement Account) is a type of savings account that provides tax benefits for retirement savings. A Roth ISA is a specific type of IRA that allows you to contribute after-tax income and then withdraw the contributions (not the earnings) tax-free. A 410(k) is an employer-sponsored retirement plan that allows employees to contribute a portion of their income to a tax-deferred retirement account. The main differences between the three are the contribution limits and the required minimum distribution rules. The contribution limits for a Roth Ira are lower than those for a 301(k), and the Roth Ira does not require minimum distributions during the contributor's lifetime. The 310 (k) requires minimum distributions to begin at age 70 1/2.

---

Our: An IRA (Individual Retirement Account) is a type of savings account that provides tax benefits for retirement savings. A Roth I RA is a specific type of IRA that allows after-tax contributions, and the money grows tax-free. A 4 01(k) is an employer-sponsored retirement plan that allows pre- tax contributions and the investments grow tax-deferred. The key differences between the three are:
* Contribution limits: The annual contribution lim it for a Roth IR A is lower than that of a 40 1 (k) .
* Eligibility: Not all employers offer 411) plans, but most employers do offer IRA plans.
* Required Minimum Distributions (R MDs): 4 O 1( k) plans require RMDs after age 70 and 1/2, while IRAs do not have RMD requirements.

In this task, you will be presented with a question, a response, and three claims. Your task is to evaluate the *faithulness* of the response. Please carefully read all elements. Initially, you will be required to rate your level of agreement with two questions. Please utilize the sliders to indicate the extent of your agreement with the statements (1 = Strongly disagree, 2 = Disagree, 3 = Neutral, 4 = Agree, 5 = Strongly agree).

Following this, you will be prompted to determine how many claims are supported by the response. Judging carefully whether each **claim** is **fully supported** by the response.

**Question: Why do muscles tend to get less sore after a work out when you work out on a regular basis?**

Response:

> Muscles tendto get lesssore aftera workout when youwork out ona regular basis because of the adaptation of the muscle fibers to the exercise. As you continue to exercise, your body becomes more efficient at repairing the damaged muscle fiber, resulting in less soreness. Additionally, the lactic acids that cause soreness are broken down quickly after aworkout, and the increased blood flow from continuous exercise helps to flush out the waste products.

Claims:

- **Your body adapts to the routines that you do on a regular basis.**
- **If you change your routine, you are more likely to be sore the next day.**
- **Regular exercise can help reduce muscle soreness.**

Your Ratings:

- 1) The **response** is fluent and cohesive (checking syntax error).

  ○————————————

- 2) The **response** is a helpful and informative answer to the question.

  ○————————————

- 3) How many **claims** are **fully supported** by the response?

  ○————————————

**Submit**

Figure 6: One example in our human evaluation experiment.

Table 11: The format for ELI5. In the context learning experiments for ELI5, each example follows a specific format. There are 2 examples in total, and for each one, it includes a question, a document, and an answer.

---

**Instruction: Write a high-quality answer for the given question using only the provided search results.**

---

**Question:** why does mining crypto use so much electricity compared to normal PC use.

---

**Document** [1](Title: How Much Electricity Does Your PC Consume? — PCMag.com): use more electricity under load than a Chromebox with a low-power CPU. Factor 3: How You Use Your PC Just because your PC is a beast with a 750-watt power supply doesn't mean it's going to use 750 watts all the time. Most PCs come with power-saving features that lower your energy usage when the computer is idle, or doing basic tasks like browsing the web. So someone mining Bitcoin or folding@home is going to use more power than someone typing up Word documents, even if they did so on the exact same PC for the same number of hours

**Document** [2](Title: Why I built a cryptocurrency mining factory in my bedroom — CCG): I found some free software online for mining Zcash and was ready to go. How the numbers stacked up The biggest cost for a crypto miner is electricity. You need to leave your computer running non-stop if you want to make maximum use of it, but this involves not only the cost of the mining itself but also the cost of keeping the computer cool. Fortunately, at that time I was living in Trinidad, which according to my research had the second-cheapest electricity in the world at just five US cents (3.7p) per kWh, compared with a typical cost of

**Document** [3](Title: Agorastoken Mining With Pc – Say it with Crypto-Currency – Bitcoins Alot): Agorastoken Mining With Pc – Crypto-Currency – Building Wealth at Each Level Thank you for coming to us in search for "Agorastoken Mining With Pc" online. The beauty of the cryptocurrencies is that scam was proved an impossibility: because of the character of the method in which it is transacted. All exchanges on a crypto-currency blockchain are irreversible. After you're paid, you get paid. This is simply not anything short-term where your visitors could challenge or demand a discounts, or use dishonest sleight of palm. Used, most dealers could be smart to utilize a transaction processor, due to the irreversible

---

**Answer:**

