# OpenReview forum: "Unlocking Anticipatory Text Generation: A Constrained Approach for Faithful Decoding with Large Language Models"
_ICLR.cc/2024/Conference — Submitted to ICLR 2024_

### Official Review · Reviewer_h9Ue · 2023-10-29

**Soundness:** 3 good
**Presentation:** 3 good
**Contribution:** 3 good
**Rating:** 5
**Confidence:** 4

**Summary:**

This paper proposes formalizing text generation as a future-constrained generation problem to minimize undesirable behaviors and enforce faithfulness to instructions. The estimation of future constraint satisfaction, accomplished using LLMs, guides the text generation process.

**Strengths:**

* The paper tackles an important problem of controlling undesirable behaviors like toxicity and hallucination in large language model text generation. This is a key challenge as models scale up.
* The method seems generic enough to handle different types of constraints like keywords, toxicity, factual correctness etc. as evidenced by the diverse experimental tasks.
* Results across the three tasks demonstrate improved constraint satisfaction and control over text generation with modest tradeoffs to fluency.
* Analysis of the proposed satisfaction score on constructed benchmarks provides useful insights.

**Weaknesses:**

* The satisfaction score estimation is currently limited and may not be robust or accurate enough for all constraints. More investigation into refining this estimation would be beneficial.
* It is not clear if the gains will sustain for very long text generation where error accumulation could occur. More analysis on larger generation tasks could help.
* There is no human evaluation of the quality and naturalness of outputs. Automatic metrics have limitations.
* The factual correctness results are quite noisy and could benefit from more tuning and robustness testing.

**Questions:**

N/A

---

> ### Author Response · Authors · 2023-11-21
>
> We would like to express our gratitude for your detailed review and feedback.
>
> ***“The satisfaction score estimation is currently limited.”***
>
> In this work, we use a scoring mechanism to identify if a constraint has not been satisfied yet to guide the future generation for LLMs. Our work highlights the demonstrated effectiveness of the current constraint score function. Additionally, the appendix, particularly in Figure 5, presents supplementary results using the score function with binary Yes/No prompts. While acknowledging the potential exploration of other scoring functions, we posit that tuned models or the utilization of smaller models with more robust or accurate future satisfaction score functions may prove more beneficial.
>
> ***“not clear if the gains will sustain for very long text generation.”***
>
> We present the model outputs of different decoding methods for the long-form QA task in Table 10. Notably, there is a reduction in error accumulation, with a significant decrease in the number of typos for the word '401k.' Additionally, the human evaluation results in Table 7 further underscore the improvements across three dimensions: Fluency, Informativeness, and Correctness.
>
> ***“Automatic metrics is limited."***
>
> In our study, we surpass traditional n-gram-based evaluation metrics by integrating constrained satisfaction measures. These include the Coverage score for CommonGen, toxicity score from the Perspective API for the toxicity task, and correctness for the ALCE benchmark. Moreover, our research extends to human evaluations across three dimensions: Fluency, Informativeness, and Correctness. The combined outcomes from these diverse evaluations consistently emphasize the benefits of constraint satisfaction, ultimately leading to faithful generation.
>
>
> ***“The factual correctness results are quite noisy”***
>
> Significantly, we observed that answers are truncated by the first newline in the ALCE's script, impacting our findings. Consequently, we present updated results on faithful question-answering generation. The new results indicate the superiority of our proposed method.

---

> > ### Author Response · Authors · 2023-11-23
> >
> > Dear Reviewer h9Ue,
> >
> > The author-reviewer discussion period will end in less than 12 hours. Please let us know if you have any concerns or questions. We would be happy to address them!

---

> ### Comment · Reviewer_h9Ue · 2023-11-30
>
> Thank you for your response, I will keep my score

---

### Official Review · Reviewer_Ht7A · 2023-10-30

**Soundness:** 2 fair
**Presentation:** 1 poor
**Contribution:** 2 fair
**Rating:** 3
**Confidence:** 4

**Summary:**

This paper points out that language model suffer from undesired behaviors such as toxicity or hallucinations and proposes a approach called future constraint satisfaction to tackle this issue. This method forces the model to take constraints into account when generating texts. The constraints here can be expressed directly in natural language. Specifically, it controls the probability of the next generated token by adding a score in the decoding stage. This score is obtained by prompt $x$ and prefix $y_{\leq}t$, using the log-likelihood. Experiments on three tasks: keyword-constrained generation, toxicity reduction, and factual correctness in question answering field show that this method can effectively improve efficiency and effectiveness compared to the baselines.

**Strengths:**

- The paper excels in its clarity and succinctness in explaining the proposed method’s core idea, namely future constraint score. The subsequent formula provides a direct method for computing this score.
- The experimental section is well-structured and comprehensive. The authors has conducted experiments on three different QA tasks, using multiple backbone models. Moreover, the impact of hyperparameters on the experimental results has also been thoroughly investigated.
- The entire paper is well-articulated, ensuring a smooth reading experience without any obscure sections.

**Weaknesses:**

- The definition of future Constraint Satisfaction is somewhat ambiguous to me. According to the formula at the bottom of page 2, your $R(y_{\leq t}, C(x))$ is used to approximate $\log p(C(x)|y _{\leq t})$, yet this is similar to the definition of $R$ provided in formula 1 on page 3. Could you please elaborate on the benefits of such a definition and why |SEP| token is added? This aspect lacks a comprehensive analysis.
- The approach considers the impact of prompt and prefix on the next token during the generation stage, with calculations only utilizing maximum likelihood estimation. The essence of future Constraint Satisfaction appears to revolve around the next token’s compatibility with the constraint. This similar idea is reflected in many controllable text generation methods, and the authors does not specifically compare these differences (only the different forms of control are mentioned in the related work section), which makes the paper seem like an incremental contribution.
- While the proposed method is relatively straightforward, it lacks robust experimental evidence to highlight its superiority over conventional decoding strategies. Moreover, the existing experimental results suggest that the improvements are rather limited.
- The paper would benefit greatly from the inclusion of a case Study and human evaluation. This would provide tangible examples of how this method improves issues like hallucinations. However, the authors solely provided results for some indicators (like BLEU) that have been proven to lack reliability.
- All the figure are not in the form of vector images, which results in distortion when the images are enlarged.

**Questions:**

na

---

> ### Author Response · Authors · 2023-11-21
>
> We would like to express our gratitude for your detailed review and feedback. We have updated our submission with more details on experiments, decoding time, human evaluation results, etc.
>
> ***“why |SEP| token is added?”***
>
> When calculating the future constraint satisfaction score R with the given prefix $y_{<=t}$​ , the use of <SEP> is employed to delineate the prefix from the language description constraint C. This separation is crucial; otherwise, the logistic score on the first token of C conditioned on the prefix $y_{<=t}$  lacks meaningful interpretation. Our preliminary experiments indicate that this setting is more effective. Further details will be expounded upon in the upcoming version.
>
> ***“differences forms of work”***
>
> Indeed, there is other related work on guided generation, but the forms of control differ significantly. In our approach, we verbalize the constraint and estimate future satisfaction scores using LLMs. Our motivation stems from our belief in the expressive power of language. Several distinctions set our work apart from previous endeavors. Firstly, we do not require additional fine-tuning for tasks. Secondly, our guidance is integrated into the token-level generation process as opposed to post-generation refinement. Thirdly, we employ natural language constraints, providing flexibility to incorporate various constraints. Lastly, our approach includes self-interpretation of these partial prefix outputs, among other unique features.
>
> ***“All the figure are not in the form of vector images.”***
>
> Thank you for bringing this to our attention. We have revised all figures, and we hope the updated versions meet your expectations. Your feedback is valuable, and if you have any further questions or need additional assistance, please don't hesitate to let us know.
>
> ***“robust experimental evidence, one would expect more significant improvements.”***
>
> We have achieved notable results, particularly in the context of CommonGen, with substantial improvements in coverage scores (e.g., Falcon-7B-Instruct: 88.7% -> 93.3%, LLaMa-2-13B-Chat: 93.6% -> 95.2%, Falcon-40B-Instruct: 88.7% -> 97.6%). Furthermore, we provide updated results on QA faithful generation on LLaMa-2-13B-Chat, addressing the issue of answer truncation caused by the first newline in the ALCE’s script. In addition to quantitative outcomes, human evaluations confirm superior improvements across three dimensions: Fluency, Informativeness, and Correctness. We believe that future gains can be achieved with a more robust or accurate future-constrained score, potentially through the refinement or tuning of models.
>
>
> ***“What is the tradeoff between time complexity and generated text quality?”***
>
> Our method incurs a linear slowdown of approximately $k$ times, primarily attributable to the overhead associated with computing future satisfaction scores on candidates. A detailed breakdown of decoding times for each example in our experiments is provided in Table 6.
>
> It is worthwhile to note that the increase in decoding time is a reasonable trade-off for achieving faithful generation. To mitigate this, several strategies can be employed to maintain generation quality while reducing time, such as selecting a smaller $k$ and opting for smaller yet finely tuned LLMs capable of efficiently computing the future constraint satisfaction score $R(\y_{<=t}, C(\x))$.

---

> > ### Author Response · Authors · 2023-11-23
> >
> > Dear Reviewer Ht7A,
> >
> > The author-reviewer discussion period will end in less than 12 hours. Please let us know if you have any concerns or questions. We would be happy to address them!

---

### Official Review · Reviewer_eM49 · 2023-11-01

**Soundness:** 2 fair
**Presentation:** 1 poor
**Contribution:** 4 excellent
**Rating:** 3
**Confidence:** 3

**Summary:**

The paper proposes future-constrained generation as a way to improve faithful decoding with large language models. This essentially introduces, in the beam search, a function of both the generated sequence (at a certain time step) and the future constraint also in the form of a natural language (e.g., "the sentence will have these concepts: run team field drill"). The function is implemented as the likelihood of generating the concatenation of both sequences using a pretrained language model. The paper shows multiple empirical results showing that the method is effective in following the constraints, even improving over a larger language model.

**Strengths:**

* The use of future constraints is interesting and intuitive since they act as (self-)evaluation, ensuring that the model is still following the constraints.

* The method is quite flexible in terms of the constraints that can be put in.

**Weaknesses:**

* While the method has been empirically shown to better perform than baselines in terms of n-gram overlap and correctness, there are other dimensions that are not reported. Firstly, since the method essentially introduces a call to a language model for each beam and for each timestep in the beam search, we expect that the decoding time is slow. How much is the tradeoff between this and "text quality"? Secondly, evaluation metrics based on n-gram overlap are not usually good rankers when the models are already very strong (which in this case they are since they are based on LLMs). Human evaluation should have been conducted. Thirdly, the authors used ALCE as a benchmark, however they did not evaluate on the QAMPARI dataset which is also part of ALCE. Finally, since the focus is on "faithful decoding", the paper should have focused on those evals as well (and not on metrics based on n-gram overlap).

* Parts of the paper are difficult to understand. For example, since there is no mention of how the experiments are set up, it was very difficult to comprehend what the authors wanted to convey in Figure 2 since it showed a bunch of previously unintroduced models and mentioned terms not defined (e.g., Ranking accuracy). This issue is repeated in Figures 4 and 5. I think overall the paper needs proofreading.

* Overall, the results do not seem convincing as most improvements (focusing on correctness which is mostly related to faithful decoding) are marginal and sometimes fail to improve over a greedy/beam search baseline. Given the complexity of the method, one would expect more significant improvements.

**Questions:**

* What is the tradeoff between time complexity and generated text quality?

---

> ### Author Response · Authors · 2023-11-21
>
> We extend our sincere appreciation for your thorough review and constructive feedback. In response, we have enhanced our submission by including additional details on experiments, decoding time, human evaluation results, and other relevant aspects. Thank you for your valuable insights.
>
> ***“Decoding time, tradeoff between time complexity and generated text quality ”***
>
> Our method incurs a linear slowdown of approximately $k$ times, primarily attributable to the overhead associated with computing future satisfaction scores on candidates. A detailed breakdown of decoding times for each example in our experiments is provided in Table 6.
>
> It is worthwhile to note that the increase in decoding time is a reasonable trade-off for achieving faithful generation. To mitigate this, several strategies can be employed to maintain generation quality while reducing time, such as selecting a smaller $k$ and opting for smaller yet finely tuned LLMs capable of efficiently computing the future constraint satisfaction score $R(\y_{<=t}, C(\vx))$.
>
> ***“Evaluation metrics”***
>
> We not only present evaluation metrics based on n-grams but also include constrained satisfaction scores, such as the Coverage score for CommonGen, toxicity score from Perspective API for the toxicity task, and correctness for the ALCE benchmark. Additionally, illustrative outputs are showcased in Table 9 and Table 10. Our study encompasses human evaluations across three dimensions: Fluency, Informativeness, and Correctness. Collectively, these findings highlight the benefits of constraint satisfaction, contributing to faithful generation.
>
> ***“Lack experiments setup details”***
>
> We update the experiment setup and add more details on these models in Section A.1.

---

> > ### Author Response · Authors · 2023-11-23
> >
> > Dear Reviewer eM49,
> >
> > The author-reviewer discussion period will end in less than 12 hours. Please let us know if you have any concerns or questions. We would be happy to address them!

---

> > ### Comment · Reviewer_eM49 · 2023-11-23
> >
> > Thanks for the response. I still feel that the evaluation lack focus on faithfulness whilst introducing a method to improve such. All datasets should have been measured with metrics that measure "faithfulness". An example of such metrics is one that checks entailment of output given the input as context.

---

### Official Review · Reviewer_cGVr · 2023-11-01

**Soundness:** 3 good
**Presentation:** 3 good
**Contribution:** 2 fair
**Rating:** 6
**Confidence:** 3

**Summary:**

This work define a decoding staretgy where the future tokens are constrained based on some lexical constraints which are defined in the prompt. The idea is to have generation that remain faithuful to the prompt and do not violate lexical constraint defined in the prompt. The authors define a novel LM scoring mechanism to identify if a constraint has not been satisfied yet to guide the future generation. The authors show performance of their methods on thress tasks: CommonGen, Toxicity reduction and Factual QA. Their method improves faithfulness to the prompt in most cases.

**Strengths:**

1.) The paper is well written and evaluation is well thought out.

**Weaknesses:**

1.) The novelty of the new constraint scoring function is fairly limited.
2.) Overall performance gains are not large and only help small sized LLMs.

**Questions:**

1.) Does the <SEP> token seperate the prompt and continuation? Is it same across all the LLMs? Is it repurposed from one of the special tokens during pre-training?
2.) Does the inference mechanism ever lead to degenerate sequences? If yes, how often does that occur?

---

> ### Author Response · Authors · 2023-11-21
>
> We would like to express our gratitude for your detailed review and feedback.
>
>  ***“The novelty of the new constraint scoring function”***
>
> In this work, we use a scoring mechanism to identify if a constraint has not been satisfied yet to guide the future generation for LLMs. Our work highlights the demonstrated effectiveness of the current constraint score function. Additionally, the appendix, particularly in Figure 5, presents supplementary results using the score function with binary Yes/No prompts. While acknowledging the potential exploration of other scoring functions, we posit that tuned models or the utilization of smaller models with more robust or accurate future satisfaction score functions may prove more beneficial.
>
>
>
> ***“Overall performance gains and only help small sized LLMs”***
>
> We have achieved notable results, particularly in the context of CommonGen, with substantial improvements in coverage scores (e.g., Falcon-7B-Instruct: 88.7% -> 93.3%, LLaMa-2-13B-Chat: 93.6% -> 95.2%, Falcon-40B-Instruct: 88.7% -> 97.6%). Furthermore, we provide updated results on QA faithful generation on LLaMa-2-13B-Chat, addressing the issue of answer truncation caused by the first newline in the ALCE’s script. In addition to quantitative outcomes, human evaluations confirm superior improvements across three dimensions: Fluency, Informativeness, and Correctness. We believe that future gains can be achieved with a more robust or accurate future-constrained score, potentially through the refinement or tuning of models.
>
>
>
> ***“Does the <SEP> token seperate the prompt and continuation?”***
>
> No, the <SEP> token is specifically employed when calculating the future constraint score to separate the prefix and verbalized constraint. It's important to note that there is no <SEP> token used to separate the prompt and continuation.
>
>
>
> ***“Does the inference mechanism ever lead to degenerate sequences? If yes, how often does that occur?”***
>
> We did not observe any degenerate sequences. The hyperparameters $k$ and $\lambda$ in Formula (2) can be adjusted to control the generation behavior.

---

> > ### Author Response · Authors · 2023-11-23
> >
> > Dear Reviewer cGVr,
> >
> > The author-reviewer discussion period will end in less than 12 hours. Please let us know if you have any concerns or questions. We would be happy to address them!

---

> ### Comment · Reviewer_cGVr · 2023-11-29
> **acknowledgement**
>
> Thank you for answering my questions.

---

### Meta-Review · Area_Chair_TEMi · 2023-12-05

**Metareview:**

This paper is concerned with constrained text generation to help mitigate issues such as hallucination and toxicity. It proposes "future-constrained generation," which modifies beam search by adding a likelihood function of both the generated sequence and a future constraint also represented as text. This method is flexible enough to handle a variety of tasks and shows improvements in toxicity reduction and question answering. However, this method is quite computationally expensive as it essentially introduces a call to a language model for each beam and for each timestep in the beam search, causing a slowdown of about 3x on CommonGen and ELI5. After considering the authors' comments, most of the reviewers are in favor of rejecting the paper due to the following concerns:

* Limited novelty: Several reviewers raised concerns about the limited novelty of this work. While the authors properly acknowledge other work on guided generation in the related work section, the reviewers considered the paper to offer mainly incremental contributions.
* Mixed empirical results: The results do not seem convincing and sometimes fail to improve over the beam search baseline. Given the added complexity and significant slowdown, one would expect more significant improvements.
* Lack of faithfulness evaluation: There appears to be a gap between the goal of the paper (faithful decoding) and what is being evaluated (correctness metrics). As reviewer eM49 asked, what happens if the response has answered the question correctly but also contains hallucinated information?

In light of these concerns, I recommend rejecting this paper.

**Justification For Why Not Higher Score:**

Limited novelty and unconvincing empirical contributions.

**Justification For Why Not Lower Score:**

N/A

---

### Decision · Program_Chairs · 2024-01-16

Reject